# Environmental Pollution Liability Insurance and Corporate Performance: Evidence from China in the Perspective of Green Development

**DOI:** 10.3390/ijerph191912089

**Published:** 2022-09-24

**Authors:** Runze Yang, Ruigang Zhang

**Affiliations:** School of Economics, Guangxi University, Nanning 530004, China

**Keywords:** environmental pollution liability insurance, corporate social responsibility, corporate performance, green development, risk management

## Abstract

Environmental pollution is an inevitable primary responsibility in the production and management of enterprises, and it is the most severe challenge to achieving green production and sustainable development. Environmental pollution liability insurance (EPLI) can transfer corporate pollution liability to insurance companies, which affects corporate performance to a certain extent. However, the influencing factors of enterprise performance are complex, and EPLI also involves multiple subjects, so the impact of EPLI on enterprise performance is also complex. At first, this paper analyzes the possible relationship between EPLI and corporate performance based on the existing literature; subsequently, based on the list of EPLI-insured companies in 2014 and 2015 published by China’s environmental protection department as a sample, this paper uses a fixed-effects model to conduct an empirical analysis, and the mediating role of corporate social responsibility (CSR) was then examined; finally, heterogeneity analysis of the initial conclusions was conducted. The following conclusions are drawn: firstly, there is a significant negative correlation between EPLI and corporate performance. Secondly, CSR played a mediating role in the effect of EPLI on corporate performance; that is, EPLI inhibited the rise of corporate performance by affecting CSR. Thirdly, the impact of EPLI on corporate performance is heterogeneous in terms of equity nature, corporate pollution level and marketization degree. The results of this paper enrich the economic impact theory of EPLI and have specific practical value for enterprise management and policymakers in the background of the green economy.

## 1. Introduction

Since the industrial revolution, western countries have gradually achieved material prosperity, but with it have come the destruction of the environment. The awareness of environmental protection has sprouted in industrialized countries, and people have slowly realized that although the natural process of the earth will slow down the increase of pollution, it cannot offset the environmental pollution burden caused by human activities [1,2]. Examples of this include population explosion, rapid depletion of resources, and increasing industrialization and urbanization. Since China’s Reform and Opening-up put forward a development guiding principle with economic construction as the top priority, China has become the second largest economy in the world. However, the rapid growth of China’s GDP has come at the expense of the environment; after years of focusing on economic development and neglecting environmental protection, China’s ecological environment can no longer be ignored. Recently, the Chinese government has increasingly mentioned a green development concept that considers economic and environmental benefits. China’s current imperfect handling and feedback mechanism for ecological pollution has brought many problems. On the one hand, there is a long tail and massive amount of environmental liability compensation, and on the other hand, there has been irreversible damage to the air, land, water quality, and human body [3]. In this case, small and medium-sized companies that are required to bear the responsibility for pollution will face a huge burden. Even for a large company that can afford it, high litigation and claims costs would seriously affect the daily operations and innovation. Since the 21st century, many countries have begun implementing effective policies to urge enterprises to take care of the environment and reduce the risk of pollution. A standard method is to jointly formulate policies with insurance associations to require or even force corporations that may have environmental pollution risks to purchase EPLI as part of a new market-based approach to environmental risk management [4]. Due to its relatively limited coverage, high premium rates, low loss rates, and lack of legal backing and specialized capabilities [5], EPLI suffers a “Best Game No One Played” dilemma and EPLI’s development in China has stalled [6].

The findings of this study can be summarized in the following three aspects. First of all, the future developmental logic of human beings must be based on the premise of environmental protection through the development of technology and energy technology iterations to achieve industrial upgrading and improve the green governance system [7,8]. The direction of industrial upgrading and development has always been highly valued by Chinese national strategy [9], so it is necessary to continue studying EPLI, which is regarded as a new green governance method. Secondly, China’s EPLI has appeared for more than 30 years. Compared with developed countries, its application degree is still in its infancy. Although there are urgent practical needs, the absence of legislation and the failure of supporting policies [6], as well as various economic reasons, means that the promotion of EPLI in various regions encounters different problems. Even though scholars have attempted some interdisciplinary approaches to help EPLI advance the range of its applications [10,11,12], they are still rarely used in practice. On the whole, the rise of EPLI has not met expectations. The progress in most promoted areas is relatively slow, and the sites that have not been promoted are still passively watching. It is strategically vital to find out its problems in the market promotion. Thirdly, for the literature research on EPLI in China, most of the existing literature research on EPLI is in the stage of theoretical analysis [13], most of which is discussion of legislation issues [14]. The few empirical studies are mainly aimed at financing costs [15], environmental management effects [16], and other issues, and there are few empirical studies on the green governance effects and eventual economic effects of EPLI. The supplementary discussion on the impact of EPLI on corporate performance is of reference value for corporate decision making and local policy formulation. The primary purpose of this study is to find out how EPLI for environmental issues affects corporate performance in the background of corporate green economic development. At the same time, as a corporate management tool for green growth, EPLI’s supervisory attributes and resulting green governance effects are of great significance to the direction of future green policies and the development prospects of other types of liability insurance.

This paper firstly consults a large number of literatures on EPLI and corporate performance, summarizes the research ideas, methods and conclusions of scholars, and forms the research hypothesis and empirical research path. This paper’s primary work and marginal contribution are to study the impact of EPLI on corporate performance and the mediating effect of CSR and heterogeneity analysis. The steps are as follows: the first is to measure and verify the overall impact of EPLI on corporate performance through the fixed effect model and robustness test, where it is found that EPLI has a specific inhibitory effect on corporate performance. The second is to study the relationship between EPLI’s insurance and CSR evaluation. On the one hand, EPLI and CSR evaluation are negatively correlated; on the other hand, CSR has played a specific mediating role in the impact of EPLI on corporate performance. The third is to study the heterogeneity of the effects of EPLI on corporate performance. Among them, for corporations with different property ownership, the impact of EPLI on state-owned corporations is lower than that of non-state-owned corporations, and it can be seen from the empirical test that the difference is relatively significant. Subsequently, the impact of EPLI on heavily polluting corporations is significant, but it does not show substantial adverse effects on non-heavy polluting corporations. Third, EPLI has a considerable impact on corporations in the eastern and western regions, which are the two poles of Chinese marketization, but not significant on insurance corporations in the middle region, whose marketization condition is more complex. From the perspective of influence degree, EPLI has a higher impact on the western region than the eastern region, which shows that there is indeed heterogeneity in the effects of EPLI on corporate performance with different types.

The rest of this paper is structured as follows (see Figure 1 for the logical framework): the second part is a literature review. The third part is the econometric test of the impact of EPLI on corporate performance. The purpose is to empirically test the effect and significance of the impact of EPLI on corporate performance through sample data, econometric models and robustness testing. The fourth part is the mechanism analysis of the impact of EPLI on corporate performance, considering the mediating effect of CSR evaluation. The fifth part is the heterogeneity analysis of corporate performance by EPLI. According to the ownership of corporate property, different pollution levels and marketization degree, the influence degree and significance of EPLI on corporate performance in different samples are studied. The sixth part draws the basic conclusions.

## 2. Literature Review

Liability insurance plays several critical roles in corporate management. From the perspective of corporate performance, corporate risk-taking means the risk choice of corporate governance and shareholders when making decisions. It reflects that enterprise managers integrate various internal and external factors to analyze and select uncertain capital investment projects. It is also the strategic trend of enterprises to strive for competitive advantages and high returns [17]. Increasing the degree of risk-taking can optimize the efficiency of capital allocation [18] and improve corporate performance [19], which is the driving force for sustainable economic growth and sustainable corporate development [20,21]. Since information asymmetry in China’s financial market is more severe than in Western countries, and the cost of entrusting environmental pollution liability control via insurance accounts for a higher proportion of working capital. The enterprise value would be affected if the degree of risk-taking is reduced and the conservative production and operation mode is adopted [20]. The importance of liability insurance for enterprises is mainly reflected in the following points: firstly, it can alleviate the distortion of industrial structure caused by judgment-proof problems [22,23]. Debtors can file for bankruptcy in numerous legal systems to eliminate insolvent debts. The problem caused by this is that small corporations are willing to take liability risks that exceed their assets to increase output at lower average production costs and increase their market share [24], forcing industry leaders to conduct vicious competition [25,26]. However, insurance can cover the risk of bankruptcy and ensure that the corporations can afford the loss. Small corporations need to consider the difficulty of filing for bankruptcy when increasing production [27]. Secondly, insurance can create incentives for corporations to reduce their own risks. After insurance participation in business operations, the bureaucratic management phenomenon in government supervision can be replaced with market-based incentive measures [28]. Adding insurance gives insurance companies access to detailed data on historical corporate liability events and helps actuaries develop detailed pricing. Since premium depends on expected losses, it increases the enterprise’s incentive to maintain a good history. From a social perspective, insurance is a “general technology to rationalize society” in promoting corporations by improve green supply chain management [29]. Thirdly, insurance can give insurance companies the right to monitor in the form of contracts, reducing the frequency of environmental pollution losses. Since insurance pricing occurs before the next accident, corporations could be motivated to raise their safety standards and seek the most significant possible reduction in insurance premiums to conduct a cost–benefit analysis and implement preventive measures [30].

The previous literature has shown that EPLI affects businesses in several ways. Environmental pollution occurs in the lifecycle of modern industries, including production, transportation, consumer use and waste disposal. Public attention has increased the need for environmental governance, and the resulting regulations have increased the risk of corporate bankruptcy [31]. Dropulić and Cular (2019) [32] believe that EPLI can improve the disclosure quality of corporate environmental information, and the supervision of local governments shows a “complementary effect”. Similar studies have also assessed the impact of EPLI on corporate pollution, the likelihood of environmental hazards and corporate environmental performance [25,33]. Their findings are that EPLI participation can improve the ecological outcomes of polluting corporations. However, some views believe in a premise that only if insurance corporations do not pay for corporate bankruptcy due to environmental pollution, EPLI will enhance environmental protection efforts, and the result of insurance is reduced production [34]. This suggests that, although insurance can urge corporations to appear environmentally friendly, the actual economic utility may not be as expected because they cannot achieve the desired output of corporations. In practice, the information asymmetry between insurance companies and corporations can be severe, and insurance can easily become a tool used by corporations to transfer responsibility. When the management of corporations take compensation as their motive for insurance, they may take some “masking effect” behaviors to justify their “risk reduction efforts” to obtain compensation [35]. If it is challenging to monitor corporations’ contamination and cover-up, insurers will take on greater payout risk, creating a moral hazard problem [24]. Some scholars have also proposed the promotion model of EPLI, such as the industry co-insurance [36] and the establishment of securitization products [11]. However, in the end, it may not alleviate the problem of enterprise performance, so it is not easy to really improve the stagnation of EPLI promotion.

Many scholars believe that a multi-layered relationship exists between CSR and environmental pollution, affecting corporate performance from various perspectives, such as the heterogeneity of industry categories or the lag effect [12,37]. With the continuous improvement of corporate governance structure, CSR has gradually attracted the attention of academia. In the early literature, that representative of neoliberal economists, Friedman and others, believed that the only social responsibility that corporations need to undertake was the interests of shareholders [34]. However, more and more scholars believe that CSR also includes multiple responsibilities to consumers, communities and the environment [38,39]. The CSR score of the Hexun survey used in this paper combines five categories, including responsibility to shareholders (30%), responsibility to employees (15%), responsibility to suppliers, customers and consumers (15%), environmental responsibility (20%), and social contribution responsibility (20%). Among them, environmental responsibility is scored by combining five dimensions: corporate environmental protection awareness, environmental management system certification, environmental protection investment amount, pollutant discharge type and energy-saving type, and the score is related to the environmental performance of corporations [40]. Although in the corporations participating in the EPLI, insurers may play several roles under contractual deductibles, such as the review and supervision of management, production, operations and risk control. There are also obligations to urge enterprises to protect the environment and prevent pollution. But in the end, the impact on corporations has little effect [13], and it will expose the behavior of corporations to transfer responsibility for environmental pollution. According to stakeholder-related theory and social contract theory, corporate management needs to maintain a principal–agent relationship formed by multiple contracts, and the execution effect of the contract directly affects the CSR evaluation of corporations [41]. However, the reputation increase brought by CSR has no obvious impact on financial value and performance, so management may be reluctant to assume social responsibility during non-crisis periods [42]. Although the government will promulgate various laws and regulations to promote corporations to implement CSR, the initiative of corporations mainly comes from corporate governance goals and strategies aimed at maximizing profits.

The investment into EPLI will help corporations avoid high liability and compensation and increase the expectation of operating at a profit, much as corporate management and shareholders will reach a better Pareto insurance decision and have the opportunity to seek government subsidies to achieve the best multi-party game strategy [43], but then there will be severe adverse selection and moral hazard problems. Specifically, first of all, due to China’s gradual loose monetary policy in recent years, corporations have increased access to financing [44], and at the same time, corporations are also required to increase profit margins and speed up producing on and operating cycles to maximize profits. With the accumulation of market monopoly, corporations have less incentive to improve the market through innovation [45], and it is easier to choose between taking risks and increasing production in the form of expanding scale. Indeed, the lowering of entry barriers for manufacturing industries has intensified competitive pressures [46]. Secondly, according to the perfect competition model, the production and operation of corporations under tremendous competitive pressure will either choose to pollute but make a profit, or strive for compensation or no pollution to obtain a small profit [47]; therefore, corporations would choose the former in the game of high premium investment. In the end, there may be a phenomenon in which bad money drives out good money in any area where EPLI exists: all corporations that can transfer pollution liability and obtain higher profits by taking out environmental liability insurance choose to insure. The increase in the loss ratio will increase EPLI premiums, crowding out corporations with less serious pollution, and further increasing the loss ratio and premiums. After such an upward spiral, in the end, only corporations with poor CSR evaluations are willing to take the initiative to choose insurance. A similar situation also occurs in other types of liability insurance, such as directors’ and officers’ liability insurance (D&O), which has been discussed frequently in recent years, and its intervention will instead contribute to the risk of corporate litigation [48]. Finally, because of corporations’ CSR disclosure, third-party audits and the disclosure of professional information platforms, corporations with serious pollution and EPLI will receive lower CSR scores because of their irresponsibility to multiple parties and poor environmental performance.

The above studies provided reference experience and new ideas for this research and related topics, and some Chinese viewpoints provide a reference value for the research route of this paper. For example, the study of the economic significance of insurance, of environmental pollution performance, of environmental information gaps, and of financing constraints are helpful to this paper’s methodology. Although they do not cover the research topic and purpose of this paper, they provide a reference for the research route of this paper and are listed in Table 1.

## 3. Research Design

### 3.1. Sample Selection and Data Sources

The purchase of EPLI does not belong to the content that the listed corporation needs to disclose, but rather that which the environmental protection department has the right to disclose, and as the EPLI of property insurance, the insurance period is usually only one year. However, the Ministry of Environmental Protection of China had only released the “Enterprise List of Environmental Pollution Liability Insurance” for 2014 and 2015; therefore, this paper selects corporations listed in Shanghai and Shenzhen A shares in 2014 and 2015 as samples, and excludes ST corporations and corporations with abnormal and incomplete data. The identification of heavily polluting corporations is mainly based on the research of Ni Juan [49], and the corporations are classified according to the “Guidelines for the Industry Classification of Listed Corporations” issued by the China Securities Regulatory Commission in 2012. In addition, 16 industry classifications were selected by comparing the scope of identification of heavily polluting industries in the “List of Industry Classification Management of Listed Corporations for Environmental Protection Verification” (Environmental Pollution Office Letter [2008] No. 373) issued by the Ministry of Environmental Protection in 2008 (Industry code: C26, C28, C27, C32, B09, B06, D44, C19, C25, B07, C17, CC22, C33, C30, C31, B08). The corporations insured by EPLI mainly sort and filter the data of the insured corporations by manually sorting out the “List of Environmental Pollution Liability Insurance Corporations” issued by the Ministry of Environmental Protection in 2014 and 2015. Insurance and corporate data come from the Cathay Pacific database and the official corporate statement. This paper carried out the 1% and 99% levels of tail processing for the explained variables to eliminate the influence of outliers and extreme values.

### 3.2. Variable Definitions

The explained variable of this study is corporate performance (Tobin’s Q). According to existing research, financial indicators of corporate performance are mainly divided into two types, Tobin’s Q-value A (market value/total assets) and accounting performance return on assets. Considering the significant volatility of China’s stock market, this paper uses Tobin’s Q-value A as an indicator to measure corporate performance. Referring to the studies of Wu et al. [50] and Chen et al. [34], this paper uses the decision of whether to purchase EPLI as an explanatory variable. If the insured corporations or its parent corporation is on the 2014 and 2015 “Environmental Pollution Liability Insurance Corporations List” and the contract is still in effect, it will be assigned a value of 1. Otherwise, it will be 0. The mediating variable studied in this paper is CSR. Since there is a coupling reaction between various responsibilities, this paper selects the comprehensive score as a proxy variable. Based on the extensive literature on corporate performance, this paper sets the following variables to control for their possible impact, they include level (asset–liability ratio), size (corporate size), cash (cash flow), ROA (net profit margin of total assets), tangible asset ratio (tangibility), ownership concentration (Top10) and corporate age (age). In addition, this paper further uses industry, year and region as dummy variables to control the problem of variable omission. The specific definition of each variable is shown in Table 2.

## 4. Results and Discussions

### 4.1. Descriptive Statistics

Table 3 lists the descriptive statistics of the main variables in this paper. As shown in the table, Tobin’s Q-variance of corporate performance is 15.2339, the minimum value is 0.7488, and the maximum value is 729.6293, which shows that there is a big difference between the performance of different heavily polluted corporations. The mean of EPLI-insured corporations is only 0.0752, indicating that the sample size of insured corporations is relatively tiny. In addition, the descriptive statistics of other variables are normal, and the influence of extreme values is not seen.

### 4.2. Benchmark Regression

According to the previous assumptions, to study the impact of EPLI on corporate performance, this paper designs the following model:Tobin’s Q_it_ = α_0_ + α_1_ * EPLI_it_ + α_2_ * Level + α_3_ * Size + α_4_ * Cash + α_5_ * ROA + α_6_ * Tangibility + α_7_ * Top10 + α_8_ * lnAge + α_9_ * lnAge2 + Year_t_ + Industry_i_ + ε_it_(1)
where among these values, the subscript i represents the industry, and t represents the year. ε_it_ is a random disturbance term. In this model, the main concern is the coefficient α_1_. The results determine the impact of EPLI on corporate performance. Tobin’s Q, if α_1_, is significantly greater than zero, and means that the corporations’ insured EPLI helps to improve the corporate performance of the heavily polluting corporations. Otherwise, it will reduce the corporate performance of the heavily polluting corporations.

This paper first conducts a quantitative test of Model 1 to examine the impact of EPLI on corporate performance. First, the OLS model is used as the starting point for the analysis, as shown in the results in columns (1) and (2). To further control the endogeneity problem caused by some unobserved heterogeneity, this paper additionally adopts a panel data model to deal with it. Since the Hausman test shows that the two-way fixed-effects model (FE) is preferable, and there are significant differences among different listed corporations, the two-way fixed-effects model with robust standard errors is used for regression, and the results are shown in columns (3) and (4). Special attention was paid to the magnitude and significance of the coefficients of the explanatory variable EPLI in the test. The results are shown in Table 4.

It can be seen from Table 4 that whether control variables are added or not, the influence coefficient of EPLI on corporate performance is significantly negative. According to the research of the Annual Report on the Development of Local Green Finance in China, the average premium of EPLI is relatively high, and the high premium will inhibit the enthusiasm of corporations to participate in insurance [51]. The main reasons for higher premiums may be adverse selection and uncapped liability limits. The first is adverse selection. Stiglitz (2013) [52], a Nobel Prize winner, pointed out that since policyholders transfer risks to insurance corporations through insurance contracts, policyholders will consciously reduce their awareness of risk prevention and pursue maximizing productivity. However, insurance corporations will actuarially consolidate the cost of this risk increase before insurance solves the underwriting risk caused by information asymmetry. Finally, adverse selection leads to higher premiums. In insurance litigation, claims decisions tend to favor policyholders, and a combination of these factors has led to higher premiums. High premiums will affect the profitability of corporate liquidity premiums, which would consume limited idle funds; the internal resources of corporations for investment activities are partially “squeezed out” [53], and both business performance and investment returns will be affected. The final result is that EPLI brings pressure to the corporation, reduces the operating cost of the corporation for production and circulation, and reduces the operating income of the corporation; the existence of EPLI, in turn, reduces the deterrent effect of laws and local environmental regulations on corporations, and increases potential litigation costs and pollution prevention costs. Some scholars, starting from Porter’s five forces model, found that EPLI will reduce the efficiency of technological innovation and resource management of corporations, leading to a reduction in the market competitiveness of corporations [54], and ultimately affecting corporate performance. As shown in column (4), the two-way fixed effects model verifies the above conclusions.

First, the regression coefficients can be unstable for various reasons. Regression analysis requires corporate performance to be normally distributed and sensitive to outliers. Outlier problems, collinearity problems, and heteroscedasticity problems can all lead to biased regression results. Additionally, we cannot understand the changing process of the influence trend of EPLI on corporate performance through regression analysis, whereas quantile regression can solve this problem very well. The regression results are shown in Table 5.

Second, we will address the bootstrap test. Here, we mainly replace the initial regression with the bootstrap repeated sampling method to obtain more effective estimation results. This paper uses the results of 3000 repeated sampling as the regression coefficient, and the regression results are shown in column (1) in Table 6.

The promotion of EPLI in China is strongly influenced by policy. Because EPLI’s pilot in China began in some eastern provinces and the industry’s leading corporations, the regions and industries where corporations are located are highly correlated with EPLI’s purchases. However, the location and industry of corporations of the same type are not directly related to corporate performance. Therefore, we can try to use province and industry as candidate instrumental variables, and the test results are shown in columns (2) and (3) of Table 6. The model of the response is shown below, the province and industry in Formula (2) are used as instrumental variables, the RKF test (weak identification test from Cragg–Donald Wald F statistic test) statistic is positive, and the *p*-value is 0, indicating no weak instrumental variable problem. Equation (3) is a further regression.
EPLIhat_it_ = δ_0_ + δ_1_ * Provincei + δ_2_ * Industryi + μ_it_
(2)
Tobin’s Q_it_ = α_0_ + α_1_ * EPLIhat_it_ + α_2_ * Level + α_3_ * Size + α_4_ * Cash + α_5_ * ROA + α_6_ * Tangibility + α_7_ * Top10 + α_8_ * lnAge + α_9_ * lnAge2 + Year_t_ + Industry_i_ + ε_it_(3)

Three robustness tests are carried out in this section, and the initial results are passed: from Table 5, in terms of quantiles, corporations in different positions are affected by EPLI to different degrees, but the results are all significant. From Table 6 column (1), after we use bootstrap to sample 3000 times, we see that the effect of EPLI on corporate performance is still significantly negative. From Table 6 column (2), (3), when instrumental variables are considered instead of covariation, we see that the negative impact of EPLI on firms is still significantly negative, indicating that the original conclusion is robust, and the inhibitory effect is four times that of the original conclusion. According to the study of David et al. (2010), this is due to the individual heterogeneity caused by the “local average treatment effect” [55]. Heterogeneity will be analyzed from multiple perspectives later in this paper.

### 4.3. Mediating Effect Analysis of CSR

When corporations fall into a crisis event, insurance performs the function of compensation for losses, and the moral capital formed by CSR plays a role in restoring wealth [56]. Much of the existing literature discussed the impact of CSR on corporations after crisis events [57,58]. But insurance pricing happens before the crisis, so EPLI’s and CSR’s combined effect on corporate performance should be re-examined. Literature also discussed those long-term corporations, with a high evaluation of CSR liable to reduce the risk of corporate stock price crashes [59]. Some scholars believe that corporations that disclose CSR would eventually generate a competitive disadvantage because unnecessary costs affect profits and shareholder wealth. Whether it is short-term analysis [60], long-term analysis, or market measurement [61] that measures abnormal returns, there is a significant negative correlation between CSR and corporate performance.

From a logical or economic point of view, CSR could play a mediating role. Therefore, this paper uses the stepwise regression method to verify the mediating effect of CSR. This paper selects the CSR score published by Hexun.com as an intermediary variable. The mediation effect model is as follows:CSR_it_ = β_0_ + β_1_ * EPLI_it_ + β_2_ * Level + β_3_ * Size + β_4_ * Cash + β_5_ * ROA + β_6_ * Tangibility + β_7_ * Top10 + β_8_ * lnAge + β_9_ * lnAge2 + Year_t_ + Industry_i_ + ε_it_(4)
Tobin’s Q_it_ = γ_0_ + γ_1_ * EPLI_it_ + γ_2_ * CSR_it_ + γ_3_ * Level + γ_4_ * Size + γ_5_ * Cash + γ_6_ * ROA + γ_7_ * Tangibility + γ_8_ * Top10 + γ_9_ * lnAge + γ_10_ * lnAge2 + Year_t_ + Industry_i_ + ε_it_(5)
where, α_1_ is still a measure of the impact of EPLI on corporate performance. β_1_ is a measure of the impact of EPLI on CSR. γ_1_ measures the direct effect of EPLI on corporate performance, and γ_2_ measures the mediating effect of CSR. This paper gradually verifies the significance of the regression coefficients. The meanings of other variables in the regression equation are the same as those described above. Due to space limitations, this paper will not elaborate on the principle of the mediation effect model. The regression results are shown in Table 7, where columns (1), (2) and (3) are the regression results of the mediating effect of CSR, and columns (4) and (5) are the analysis results of clustering robust standard errors in both industry and individual directions.

As can be seen from Table 7, EPLI has a significant negative effect on corporate performance, and CSR plays a mediating role and passes the robustness test. The opportunity cost of corporation’s investing in CSR may enhance a corporation’s reputation because CSR also embodies a focus on stakeholders other than shareholders, and corporate performance would be ultimately rewarded in the long run. However, it will not generate benefits immediately, because the cost of input always exists, so CSR and current corporate performance should be negatively correlated [62]. Generally speaking, the main purpose of corporations purchasing EPLI is to improve efficiency. The agency theory holds that there is an unavoidable conflict between corporate managers and shareholders in the decision making of risk taking: the management pays more attention to overall interests than the shareholders’ interests. For example, when corporations face the supervisory pressure of customers or insurance industry associations and need to adjust their operational decisions, management tends to choose a “greenwashing” strategy, that is, to use superficial or misleading disclosures to maximize the interests of the corporation and its long-term development [63], while shareholders will pursue the maximization of shareholders’ equity. China is currently in the early stage of CSR development, and corporations are not motivated enough to take the initiative to undertake CSR [64]. The behavior of corporations, whereby they perform and disclose CSR, has become a proxy tool for management to avoid risks [65]. Some corporate managements use insurance policy as a tool to transfer litigation and compensation risks and use CSR activities as a “fig leaf” for litigation risks to cover up various problems and deficiencies. When the government and CSR announce information about purchasing insurance, it is used as a management benefit management tool rather than a shareholder value tool and the way CSR is performed will ultimately remain on the surface. Low-quality CSR information disclosure cannot reduce information asymmetry but aggravates the principal–agent problem. The FE model verified the above conclusion.

### 4.4. Heterogeneity Analysis

#### 4.4.1. Heterogeneity Analysis Based on Actual Controllers

First, we will address the test for the nature of property rights. Chinese EPLI has a strong policy influence, which will seriously affect the decision making of the actual corporate controller. In practice, an imperfect legal system means more opportunities for corporations to evade regulation, and even the implementation of punishment cannot achieve incentive compatibility [66]. The frequency of public incidents in developing countries where corporations use different standards for illegal production and operation is very high, and it is difficult to supervise them effectively; this kind of omission allows corporations to have more flukes to increase production and thus increase risks. Third, even if there is an unavoidable payment, EPLI will still be responsible for the cost. Although the number of environmental compensation lawsuits in China is increasing [67], considering the fairness and instrumental role of insurance, EPLI does not set a maximum compensation limit, and it is still based on the insurance amount [54]. This paper takes property ownership rights as the dummy variable ownership and assigns the value of 1 to the corporation whose actual controller is the government, and the value of 0 to the corporation for whom the government is not the actual controller. The model is shown in the Formula (6). The regression results are shown in column (1) in Table 8. The model is as follows, in addition.
Q_it_ = α_0_ + α_1_ * EPLI_it_ + α_2_ * Ownership + α_3_ * Level + α_4_ * Size + α_5_ * Cash + α_6_ * ROA + α_7_ * Tangibility + α_8_ * Top10 + α_9_ * lnAge + α_10_ * lnAge2 + Year_t_ + Industry_i_ + ε_it_(6)

A group regression was also performed on the actual controllers, and the results are shown in columns (2) and (3). The regression results of state-owned enterprises are in column (2), and those of non-state-owned enterprises are in column (3).

As can be seen from Table 8, the impact of EPLI on corporate performance does vary significantly in terms of the actual ownership of property rights. First, after controlling for the ownership of property rights as a dummy variable, the preliminary regression result is still that EPLI has a negative impact, and the negative impact is smaller for corporations whose actual owner is the state, but the negative impact is more profound for non-state-owned corporations. There may be two reasons: firstly, state-owned corporations have significant capital, and insurance premiums have a relatively low occupancy rate of working capital and limited impact on business operations; secondly, state-owned corporations themselves have a good corporate image and reputation, and the purchase of EPLI will not become public information. Small businesses are more sensitive to the impact of EPLI.

#### 4.4.2. Heterogeneity Analysis Based on Pollution Degree

Since heavily polluting and non-heavily polluting corporations differ greatly in environmental aspects, this paper further studies the impact of environmental liability insurance on the performance of corporations with different pollution degrees. On the one hand, heavy-polluting corporations should pay more attention to EPLI, while non-heavy polluting corporations usually do not show a concern for compensation of environmental pollution liability; on the other hand, China has a stricter supervision and management mechanism for heavy-polluting corporations, which makes heavy-polluting corporations need insurance to deal with various lawsuits and compensation. In this paper, a group regression is carried out on the heavily polluting corporations and the non-heavy-polluting corporations, and the results are shown in Table 9. column (1) is listed as a heavy pollution industry corporation, and column (2) is listed as a non-heavy pollution industry corporation.

As seen from Table 9, only the purchase of EPLI by heavy-polluting corporations significantly impacts corporate performance. The possible reason is that heavy pollution corporations have a higher risk of environmental pollution liability, and face higher compensation when risk accidents occur. Although the high premium of EPLI will bring various financial and operational difficulties to corporations [68], EPLI insurance can not only transfer part of the compensation liability, respond to the policy requirements of the local government, but also disclose it as part of CSR to improve the corporations of moral capital and goodwill [69]. Therefore, driven by various factors, there is a certain adverse selection problem in EPLI’s insurance behavior. A higher pollution degree leads to a higher expected utility of corporations’ EPLI insurance. For non-heavy pollution corporations, the purpose of purchasing environmental protection liability insurance is less to transfer the risk of environmental pollution liability. Although the insurance cost is relatively high, it can alleviate the financing constraints of the enterprise [70] and improve its corporate performance.

#### 4.4.3. Heterogeneity Analysis Based on Marketization Degree

In 2008, China began to pilot EPLI formally. At the beginning of the pilot, there were very few samples, and it was only piloted in some provinces and cities published in the “Guiding Opinions on Environmental Pollution Liability Insurance.” Subsequently, the scope of the pilot project was gradually expanded, and the nationwide promotion was not carried out until 2013. This has led to significant differences in the promotion of EPLI and the acceptance of EPLI among different regions. It can be seen from the list of EPLI-insured corporations published by the Ministry of Environmental Protection in 2014 and 2015 that the number of insured corporations in Jiangsu Province, which has a high degree of promotion, was 1932 and 2213, respectively for two consecutive years, while the lowest in Xinjiang was only 1 in 2014, compared to 2 in 2015. To verify the impact of the marketization process, this paper conducts a group regression according to the interpretation of the National Bureau of Statistics on the division of the east, the center and the west. Among them, the eastern region has the highest degree of marketization, the western region is the worst, and the central region has an intermediate degree. The regression test of the response was carried out. The test results are shown in Table 10.

Columns (1) (2) (3) in Table 10 represent the regression results of the east, middle and west, respectively. The EPLI regression coefficient of the eastern region, which has a relatively rapid degree of marketization, is −0.337, while that of the western region reaches −0.810. Due to the varying level of marketization in the central region, the significance of the regression has diminished, but is still significant. It can be found that the degree of marketization can alleviate the negative impact of EPLI on corporate performance to a certain extent, but it will not turn losses into positive ones. The impact of EPLI on corporate performance for corporations in more market-oriented regions is lower than that in the western areas where marketization is slower. According to Rajan’s research [71], regions with a high degree of marketization have lower corporate financing costs and more investment opportunities. The promotion of EPLI is faster in regions with higher marketization, and the policy support is also higher. Local corporations will have more choices to deal with the high premiums of EPLI, which can greatly alleviate the problem of insufficient enthusiasm of corporations for insurance.

## 5. Conclusions

First, the purchase of EPLI by corporations can have a significant negative impact on corporate performance. On the one hand, EPLI purchases can affect the capital flow of corporations, resulting in less capital being available to corporations for operations. On the other hand, the supervision and control of insurance corporations brought by the purchase of EPLI, as well as the constraints of regulations, will enhance the self-prevention level of corporations, which cannot fully maximize production. Furthermore, negative effects remained significant after excluding outliers, sample size bias, and cluster hierarchical standard errors.

Second, CSR, as a score of corporate responsibility to various fields of objects, has a mediating effect on the impact of EPLI on corporate performance. The purchase of EPLI by corporations will not only hurt the evaluation of CSR but will ultimately further affect corporate performance. The impact of EPLI on corporate performance is partly through price and insurance effects, and partly through the disclosure of environmental information, which reduces CSR scores and corporate reputation.

Finally, the impact of EPLI on corporate performance is heterogeneous, reflected in the differences in property ownership, corporate pollution degree and the degree of marketization in the region. First, EPLI has a more significant impact on non-SOEs than SOEs; second, the impact on heavily polluting corporations is greater; third, regions with a high degree of regional marketization will offset the negative effects of some EPLI purchases.

The existing literature has focused on branch studies of EPLI, CSR, and firm performance, providing useful evidence for this study, but not directly reflecting their relation. The research in this paper increases the marginal contribution of EPLI’s economic effects, which may provide a reference for corporate green management decisions. However, the research content of this paper has certain limitations. First of all, since the public data on EPLI purchases include too few years, the lagging effect of EPLI cannot be studied, and the effect of long-term renewal is also worth researching; secondly, the reasons for each enterprise to purchase EPLI need to be analyzed from the perspective of cost. However, the current development of EPLI in China may lead to the fact that the terms of the insurance policy cannot be formulated according to the individual risk level but can only reflect the differences between regions and industries. Thirdly, the influence of the development of the times on the theme needs to be studied. However, due to data availability, this paper cannot cover this part of the analysis for the time being. It is hoped that the research in this paper will take these limitations into account in the future and provide valuable risk management approaches for corporate green development strategies.

Based on the research in this paper, the following insights can be obtained: first, government departments should continue to support the reform of China’s EPLI system. They should further develop and refine the division of EPLI subjects and formulate differentiated insurance cooperation models for corporations in different regions, with different property ownership and different pollution degrees, and gradually extend to other liability insurance application scenarios, to better play the social management function of insurance and promote the sustainable development of the green economy. Second, they should unswervingly adhere to the basic guiding principle of “government promotion and market operation.” Based on the existing state of supply and demand, the supply of insurance is promoted in the form of policy subsidies. Government should encourage corporations to participate in insurance and disclose environmental information governance, and to steadily resolve the embarrassing situation of “double cold supply and demand” caused by adverse selection of liability insurance. They should promote the preference for cleaner production and the commitment to environmental liability in the process of deepening the reform of corporations, so that corporations can combine internal and external governance with improving corporate performance. Third, government departments should pay more attention to and improve the construction of an ecological and environment legal system. Government should not only improve the compensatory punishment for pollution, but also make efforts to create a better atmosphere for social supervision, to stimulate the concern and enthusiasm of the general public, media, social organizations and other groups for measures to protect the environment. Finally, they should act by formulating policies to help insurance institutions maximize their professional supervision and management functions, thereby helping liability insurance to play its essential functions from multiple directions.

## Figures and Tables

**Figure 1 ijerph-19-12089-f001:**
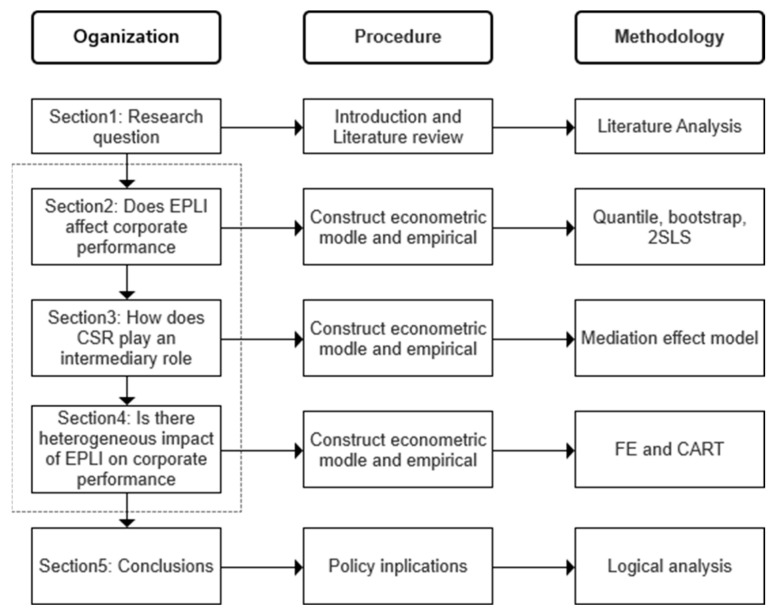
The logical framework of this paper.

**Table 1 ijerph-19-12089-t001:** Research route-related literature.

Subject	Literature
EPLI and corporate performance	[12,28,32,48]
EPLI and CSR	[24,26,30,31,33]
CSR and corporate performance	[35,40]

**Table 2 ijerph-19-12089-t002:** Variable types and definitions.

Variable Type	Variable Symbol	Variable Definitions
Explained variable	Tobin’s Q	Market Value/Total Assets, takes its natural logarithm.
Explanatory variables	EPLI	Dummy variable, takes value 1 if insured, 0 otherwise.
Mediating variable Control variable	CSR	Hexun CSR total score.
	Level	Total Liabilities/Total Assets.
	Size	The natural logarithm of the annual stock market value.
	Cash	Net cash flow from operating activities/total assets.
	ROA	Net profit/average balance of total assets.
	Tangibility	Net Fixed Assets/Total Assets.
	Top10	The total shareholding ratio of the top ten shareholders of the corporate.
	Age	Insurance year—listing year, take the natural logarithm
	Year	Year dummy variable.
	Industry	Industry dummy variable.
	Region	Region dummy variable.
	Ownership	1 for state-owned corporations and 0 for non-state-owned corporations.

**Table 3 ijerph-19-12089-t003:** Descriptive statistics.

Variable	Obs	Mean	Std. Dev.	Min	Max
Q	4814	3.06	15.23	0.75	729.60
EPLI	4814	0.08	0.26	0.00	1.00
Level	4814	0.46	0.95	0.01	63.97
Size	4810	16.69	17.89	11.71	21.28
Cash	4810	0.04	0.08	−0.89	0.88
ROA	4814	0.04	0.22	−14.59	0.67
Tangibility	4814	0.93	0.10	0.15	1.00
Top10	4810	58.19	15.65	1.32	101.20
Age	4814	16.13	5.18	4.00	47.00

**Table 4 ijerph-19-12089-t004:** The impact of EPLI on corporate performance.

	(1)	(2)	(3)	(4)
	Tobin’s Q	Tobin’s Q	Tobin’s Q	Tobin’s Q
EPLI	−0.495 ***	−0.553 ***	−0.481 ***	−0.529 ***
	(−5.00)	(−5.76)	(−5.00)	(−5.65)
Level		−1.081 ***		−0.862 ***
		(−10.61)		(−8.60)
lnSize		0.144 ***		0.000167
		(4.96)		(0.01)
Cash		1.310 ***		1.141 ***
		(3.81)		(3.40)
ROA		−4.852 ***		−3.926 ***
		(−11.05)		(−9.09)
Tangibility		−1.231 ***		−1.015 ***
		(−4.63)		(−3.92)
Top10		−0.0209 ***		−0.0195 ***
		(−12.10)		(−11.59)
lnAge		1.548 **		1.324
		(2.04)		(1.79)
lnAge2		−0.326 **		−0.278
		(−2.23)		(−1.95)
Time control	NO	NO	YES	YES
Individual control	NO	NO	YES	YES
_cons	2.634 ***	1.538	2.632 ***	3.685 ***
R-squared	0.0050	0.0684	0.0052	0.0649

Notes: **, *** stand for significant levels of 10%, 5%, and 1%, respectively, and the values in brackets are T-values.

**Table 5 ijerph-19-12089-t005:** Robustness test of quantile regression.

	(1)	(2)	(3)
	Tobin’s Q	Tobin’s Q	Tobin’s Q
	Q25	Q50	Q75
EPLI	−0.133 ***	−0.259 ***	−0.460 ***
	(−4.12)	(−8.28)	(−4.47)
Level	−1.240 ***	−2.044 ***	−2.545 ***
	(−10.86)	(−8.49)	(−3.33)
lnSize	0.0300 *	0.0998 ***	0.190 ***
	(2.30)	(4.63)	(4.92)
Cash	−0.124	−0.00367	0.876
	(−0.90)	(−0.01)	(1.13)
ROA	0.307	−0.713	−3.295 *
	(0.72)	(−1.05)	(−1.98)
Tangibility	−0.344 **	−0.562 *	−1.060
	(−3.01)	(−2.50)	(−1.73)
Top10	−0.00763 ***	−0.0137 ***	−0.0265 ***
	(−8.05)	(−10.51)	(−8.18)
lnAge	0.642 **	1.229 **	2.466 **
	(3.18)	(2.85)	(2.99)
lnAge2	−0.142 ***	−0.261 **	−0.506 **
	(−3.54)	(−3.17)	(−3.11)
Time control	YES	YES	YES
Individual control	YES	YES	YES
_cons	1.761 ***	1.383	0.734
R-squared	0.0962	0.0963	0.0696

Notes: *, **, *** stand for significant levels of 10%, 5%, and 1%, respectively, and the values in brackets are T-values.

**Table 6 ijerph-19-12089-t006:** Robustness test for bootstrap and cluster robust standard errors.

	(1)	(2)	(3)
	Tobin’s Q	Tobin’s Q	Tobin’s Q
EPLI	−0.529 ***		−2.117 ***
	(−6.65)		(−4.63)
Level	−0.862	−0.862 ***	−1.125 ***
	(−1.23)	(−8.60)	(−10.69)
lnSize	0.000167	−0.0119	0.154 ***
	(0.00)	(−0.40)	(5.13)
Cash	1.141	1.249 ***	1.517 ***
	(1.10)	(3.70)	(4.24)
ROA	−3.926 *	−3.945 ***	−5.059 ***
	(−2.37)	(−9.13)	(−11.13)
Tangibility	−1.015 **	−1.053 ***	−1.111 ***
	(−2.86)	(−4.06)	(−4.04)
Top10	−0.0195 ***	−0.0195 ***	−0.0215 ***
	(−8.58)	(−11.55)	(−12.09)
lnAge	1.324	1.457 *	1.784 *
	(1.96)	(1.96)	(2.28)
lnAge2	−0.278 *	−0.298 *	−0.377 *
	(−2.02)	(−2.09)	(−2.50)
EPLIhat		−2.176 ***	
		(−4.99)	
Time control	YES	YES	YES
Individual control	YES	YES	YES
_cons	3.685 **	3.823 ***	1.190
R-squared	0.0684	0.0664	0.0187

Notes: *, **, *** stand for significant levels of 10%, 5%, and 1%, respectively, and the values in brackets are T-values.

**Table 7 ijerph-19-12089-t007:** The mediating effect of CSR on the impact of EPLI on corporate performance.

	(1)	(2)	(3)	(4)	(5)
	Tobin’s Q	CSR	Tobin’s Q	Tobin’s Q	Tobin’s Q
EPLI	−0.529 ***	−1.967 **	−0.541 ***	−0.541 ***	−0.541 ***
	(−5.65)	(−2.65)	(−5.76)	(−7.54)	(−5.96)
CSR			−0.00486 **	−0.00486 *	−0.00486 *
			(−2.66)	(−2.51)	(−2.34)
Level	−0.862 ***	14.88 ***	−0.776 ***	−0.776 ***	−0.776 ***
	(−8.60)	(18.65)	(−7.44)	(−3.88)	(−3.73)
lnSize	0.000167	3.651 ***	0.0106	0.0106	0.0106
	(0.01)	(15.08)	(0.34)	(0.34)	(0.28)
Cash	1.141 ***	15.59 ***	1.239 ***	1.239 *	1.239
	(3.40)	(5.85)	(3.67)	(1.98)	(1.81)
ROA	−3.926 ***	69.27 ***	−3.539 ***	−3.539 ***	−3.539 ***
	(−9.09)	(20.15)	(−7.83)	(−4.18)	(−4.01)
Tangibility	−1.015 ***	1.632	−0.991 ***	−0.991 ***	−0.991 **
	(−3.92)	(0.79)	(−3.80)	(−3.72)	(−3.07)
Top10	−0.0195 ***	0.0393 **	−0.0190 ***	−0.0190 ***	−0.0190 ***
	(−11.59)	(2.93)	(−11.25)	(−10.65)	(−8.70)
lnAge	1.324	−9.707	1.293	1.293	1.293
	(1.79)	(−1.65)	(1.74)	(1.89)	(1.55)
lnAge2	−0.278	1.980	−0.270	−0.270 *	−0.270
	(−1.95)	(1.74)	(−1.89)	(−2.00)	(−1.63)
Time control	YES	YES	YES	YES	YES
Individual control	YES	YES	YES	YES	YES
_cons	3.685 ***	−37.40 ***	3.554 **	3.554 **	3.554 **
R-squared	0.0649	0.2179	0.0651	0.0651	0.0651

Notes: *, **, *** stand for significant levels of 10%, 5%, and 1%, respectively, and the values in brackets are T-values.

**Table 8 ijerph-19-12089-t008:** Heterogeneity of property ownership.

	(1)	(2)	(3)
	Tobin’s Q	Tobin’s Q	Tobin’s Q
EPLI	−0.4609 ***	−0.2437 **	−0.6172 ***
	(0.0921)	(0.1092)	(0.1386)
Level	−0.8588 ***	−2.2735 ***	−0.6589 ***
	(0.0984)	(0.1692)	(0.1332)
lnSize	0.0638 **	−0.0018	0.0970 **
	(0.0296)	(0.0347)	(0.0482)
Cash	1.0751 ***	−0.8278 *	1.3972 ***
	(0.3295)	(0.4783)	(0.4427)
ROA	−3.9873 ***	−3.4794 ***	−3.2486 ***
	(0.4241)	(0.6504)	(0.5772)
Tangibility	−0.6883 ***	0.1422	−0.8674 **
	(0.2558)	(0.3610)	(0.3448)
Top10	−0.0199 ***	−0.0159 ***	−0.0205 ***
	(0.0017)	(0.0023)	(0.0023)
lnAge	1.4514 **	0.9638	1.4303
	(0.7273)	(0.9860)	(1.0207)
lnAge2	−0.2583 *	−0.1758	−0.2412
	(0.1401)	(0.1850)	(0.1989)
Dummy variable	YES	NO	NO
Time control	YES	YES	YES
Individual control	YES	YES	YES
_cons	2.1683 **	3.0207 **	1.6699
R-squared	0.1018	0.1348	0.0649

Notes: *, **, *** stand for significant levels of 10%, 5%, and 1%, respectively, and the values in brackets are T-values.

**Table 9 ijerph-19-12089-t009:** Heterogeneity of pollution degree.

	(1)	(2)
	Tobin’s Q	Tobin’s Q
EPLI	−0.520 ***	−0.236
	(−4.42)	(−1.57)
Level	0.299 *	−2.546 ***
	(1.79)	(−16.78)
lnSize	−0.0405	0.0953 ***
	(−0.71)	(2.77)
Cash	−0.890	0.405
	(−1.44)	(0.99)
ROA	0.865	−2.696 ***
	(1.20)	(−4.56)
Tangibility	−1.524 ***	−0.306
	(−2.87)	(−1.05)
Top10	−0.0209 ***	−0.0220 ***
	(−6.90)	(−11.08)
lnAge	−3.272 **	2.462 ***
	(−2.17)	(2.96)
lnAge2	0.630 **	−0.468 ***
	(2.19)	(−2.91)
Time control	YES	YES
Individual control	YES	YES
_cons	9.827 ***	0.755
R-squared	0.0745	0.1232

Notes: *, **, *** stand for significant levels of 10%, 5%, and 1%, respectively, and the values in brackets are T-values.

**Table 10 ijerph-19-12089-t010:** Heterogeneity of marketization degree.

	(1)	(2)	(3)
	Tobin’s Q	Tobin’s Q	Tobin’s Q
EPLI	−0.337 **	−0.415 *	−0.810 ***
	(−2.85)	(−2.09)	(−3.73)
Level	−2.272 ***	−2.507 ***	−0.600 **
	(−15.01)	(−7.80)	(−2.61)
lnSize	0.0727 *	−0.0173	0.0558
	(2.18)	(−0.21)	(0.65)
Cash	0.275	−1.117	−2.754 ***
	(0.68)	(−1.20)	(−3.31)
ROA	−0.175	−3.230 *	−3.134 **
	(−0.32)	(−2.42)	(−3.14)
Tangibility	−0.698 *	−0.0560	−0.676
	(−2.34)	(−0.09)	(−0.87)
Top10	−0.0228 ***	−0.0194 ***	−0.0229 ***
	(−11.80)	(−4.62)	(−5.01)
lnAge	1.396	2.378	−3.678
	(1.68)	(1.34)	(−1.53)
lnAge2	−0.256	−0.422	0.689
	(−1.59)	(−1.22)	(1.53)
Time control	YES	YES	YES
Individual control	YES	YES	YES
_cons	2.580 *	1.958	8.953 *
R-squared	0.1277	0.1035	0.1297

Notes: *, **, *** stand for significant levels of 10%, 5%, and 1%, respectively, and the values in brackets are T-values.

## Data Availability

Not applicable.

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
