# Peer review of "Environmental Pollution Liability Insurance and Corporate Performance: Evidence from China in the Perspective of Green Development"

_ijerph, 2022, doi:10.3390/ijerph191912089_

Round 1

Reviewer 1 Report

Please find the comments in attachment.

Reviewer 2 Report

The paper entitled, ‘The impact of environmental pollution liability insurance on corporate performance from the perspective of green development’ is a much-needed topic for the current situation in China and other industrialized nations. This paper is well structured and presented to the readers. This can be accepted with minor revision.

Hope the following suggestions will improve the quality of the manuscript:

#1) Figure 1 is not really needed as this is not thesis OR review paper.

#2) Typos and Grammatical mistakes are quite all over. For example, the line 110 has the word ‘increase’ two times back-to-back. In Line 318, ‘by policy’ can be corrected as ‘by the policy’.    

Thanks

Reviewer 3 Report

Dear authors,

After reading your interesting paper, it is necessary to proofreading throughout the manuscript. There are still some questions to be addressed, see my comments below for more details:

1.Concerning about sample selection and data sources, insurance coverage list published by the Ministry of Environmental Protection in 2014 and 2015. The data is retrieved from 7 years ago and data coverage is really too old. Please update the data coverage, especially for pro-COVID-19 era.

2.In the empirical section, I was expecting a comparison with real quoted Environmental Pollution Liability Insurance, as done by Wu, W.Q.; Zhang, P.P.; Zhu, D.Y.; Jiang, X.; Jakovljevic, M. Environmental Pollution Liability Insurance of Health Risk and Corporate Environmental Performance: Evidence From China. Frontiers in Public Health 2022, 10. Why authors did not do such comparison?

3.Regarding on innovation issues, the manuscript has the same problem as the previous point 2. This manuscript is too similar to the data source as well as the research topic of the cited paper #4 of Wu et al (2022).

  4.The log-value of SIZE in Table 2. Descriptive statistics is wrong, please        correct it.

5.In line 325, what is The RKF test statistic?

6.Can you describe the specific contribution of this article to environment friendly?

Round 2

Reviewer 3 Report

Dear authors,

 There are still some questions to be addressed, see my comments below for more details: 

1.log-value of SIZE in Table 2 is still wrong. If the value is 17700000.00 then take logarithm value of SIZE must equal to 16.689 not 1.8E+07. Similarly, Std.Dev. Min and Max must collect. Because the manuscript define SIZE as the natural logarithm of the annual stock market value.

2.Please check the place of Tobin’s Q in Table 3?

3.In line 403, equation number i.e., equation (1) is correct?
